# Ranking evaluation metrics from a group-theoretic perspective

## Abstract

Confronted with the challenge of identifying the most suitable metric to validate the merits of newly proposed models, the decision-making process is anything but straightforward. Given that comparing rankings introduces its own set of formidable challenges and the likely absence of a universal metric applicable to all scenarios, the scenario does not get any better. Furthermore, metrics designed for specific contexts, such as for Recommender Systems, sometimes extend to other domains without a comprehensive grasp of their underlying mechanisms, resulting in unforeseen outcomes and potential misuses. Complicating matters further, distinct metrics may emphasize different aspects of rankings, frequently leading to seemingly contradictory comparisons of model results and hindering the trustworthiness of evaluations.

We unveil these aspects in the domain of ranking evaluation metrics. Firstly, we show instances resulting in inconsistent evaluations, sources of potential mistrust in commonly used metrics; by quantifying the frequency of such disagreements, we prove that these are common in rankings. Afterward, we conceptualize rankings using the mathematical formalism of symmetric groups detaching from possible domains where the metrics have been created; through this approach, we can rigorously and formally establish essential mathematical properties for ranking evaluation metrics, essential for a deeper comprehension of the source of inconsistent evaluations. We conclude with a discussion, connecting our theoretical analysis to the practical applications, highlighting which properties are important in each domain where rankings are commonly evaluated. In conclusion, our analysis sheds light on ranking evaluation metrics, highlighting that inconsistent evaluations should not be seen as a source of mistrust but as the need to carefully choose how to evaluate our models in the future.

## 1 Introduction

Evaluating methods is fundamental in any machine learning field, but it is not straightforward finding an *appropriate* evaluation metric that accurately assesses a method's strengths without unfairly biasing comparisons to other approaches. Comparing rankings is particularly challenging: inconsistencies often appear in the produced evaluations, and, notably, inconsistencies diminish users' trust in the methods and evaluations.

Rankings pop up in several domains, from Recommender Systems (RS) and Information Retrieval (IR) techniques Adomavicius & Tuzhilin (2005); Schütze et al. (2008), to feature ranking and selection approaches Khaire & Dhanalakshmi (2022) as well as in (fair) rank aggregation methods Lin (2010). They are looked at as "interpretable" and easy to understand by humans; as an example, let us think about a scoring system that represents job applicants' characteristics, where a high score is an indicator of a better fit for the position. However, evaluating rankings is not at all that simple, and contradictory evaluations are commonplace. Furthermore, several metrics are appropriately created to evaluate, for example, Recommender Systems or rank aggregation approaches; however, the same metrics are often used in other contexts without a sufficient understanding of their evaluation.

First, we show the existence of inconsistencies among most pairs of metrics; given the frequency of inconsistencies' occurrences, evaluations of methods involving rankings are often not trustworthy. We then introduce a list of desirable theoretical properties for ranking evaluation metrics and provide the mathematical framework underpinning each of them; rankings metrics can be easily transferred to functions on mathematical groups, specifically on symmetric groups, thus allowing us to detach from specific machine learning domains. The choice is dictated by the fact that they represent the most general mathematical structure on which we could represent rankings. Profiting from a strong mathematical theoretical interface, we look to answer the question *which mathematical properties are essential for evaluating rankings?* Thus, while most of the existing literature predominantly confines itself rather to narrow, highly specific contexts, our work provides a theoretical framework beyond specific contexts of application. We further assert that almost none of the metrics qualifies as a mathematical distance, as fundamental characteristics are not satisfied. Although symmetric groups offer a broad generalization, we remain mindful of the specific contexts in which the metrics were developed. We highlight the contexts where these properties are particularly valuable in a conclusive discussion.

## 2 Related work

The literature on ranking evaluation metrics is mostly highly context-specific. Particularly developed for RS and IR evaluation, we find several works exploring the relationships among the metrics Valcarce et al. (2018); Gunawardana et al. (2012); Silveira et al. (2019). Herlocker et al. (2004) proposes a theoretical division of the metrics for comparing collaborative filtering RS, while Liu et al. (2009) describes most of the metrics typically used for RS and IR techniques. Järvelin & Kekäläinen (2002) presents various metrics based on cumulative gain, highlighting their main advantages and drawbacks, Hoyt et al. (2022) introduces a theoretical foundation for rank-based evaluation metrics, and Amigó et al. (2018) defines a set of properties for IR metrics and shows that none of the existing ones satisfy all the properties proposed. Other works focus on metrics for RS and their intrinsic properties or on ranking metrics for the top-$n$ recommendations Buckley & Voorhees (2004); Valcarce et al. (2020). Real-world applications such as the design of strategies based on customers' feedback and allocation of priorities in R&D extended the interest in defining distances among rankings where the focus of the problem statement is *rank aggregation* Dwork et al. (2001); Sculley (2007). Examples of similarly scoped works are Cook et al. (1986); Fligner & Verducci (1986). An interesting generalization work is presented by Diaconis (1988), that focuses on six metrics on symmetric groups; we find among them, *Kendall's $\tau$* and *Spearmann's $\rho$* while the other considered metrics are rather uncommon in machine learning. The work studies them from a statistical and theoretical perspective and defines some properties, among which the *interpretability, tractability, sensitivity*, i.e., the ability of one metric to range among the available counter-domain, and *theoretical availability*. In detail, the *interpretability* defined in Diaconis (1988) discussed whether the metrics measure something humanly tangible, the *tractability* studies the so-called computational complexity in computer science, and the *theoretical availability* asks whether a metric is studied and used enough in the state-of-the-art works. We will reintroduce one of the properties in Diaconis (1988), e.g., the *right-invariance*, in our work into our *Robustness property*.

The interest in fair and trusted choices of evaluation metrics grows fast in computer science. Tamm et al. (2021) is an example where some of the ranking evaluation metrics are harshly criticized for their comparisons' reliability. In other domains, the state-of-the-art literature started defining essential properties for metrics Gösgens et al. (2021a;b); we aim to fill the gap for ranking evaluation metrics.

## 3 Metrics

In the state-of-the-art literature, we find various metrics meant to evaluate ranking in specific contexts. This is the case for most offline Recommender Systems metrics, some evaluation metrics for prediction models, and rank aggregation approaches. Some of these metrics started spreading to other domains following the need to evaluate rankings. However, this is not always a good idea as the domain defines the evaluation's exigencies. In our work, we consider metrics evaluating full rankings that can be easily transferred to adjacent domains and cut out from the analysis all those metrics that require context-specific information, e.g., diversity in

| Ranking aware metrics | **nDCG**, **DCG**, meanRank, GMR, MRR |
|---|---|
| Metrics assigning equal importance to each position | *SMAPE*, *MAPE*, *MAE*, *RMSE*, *MSE*, $R^2$ *score*, NDPM, Spearmann $\rho$, Kendall's $\tau$ |
| Set based metrics | markedness, PT, recall, LR+, Jaccard index, F1 score, FDR, accuracy, MCC, TNR, fallout, FNR, LR- informedness, NPV, FOR, BA, FM, precision |

Table 1: List of considered metrics; bold, italic, underlined, and plain text indicate **CGB**, *EB*, CMB, and CB metrics. Other metrics are blue color-coded.

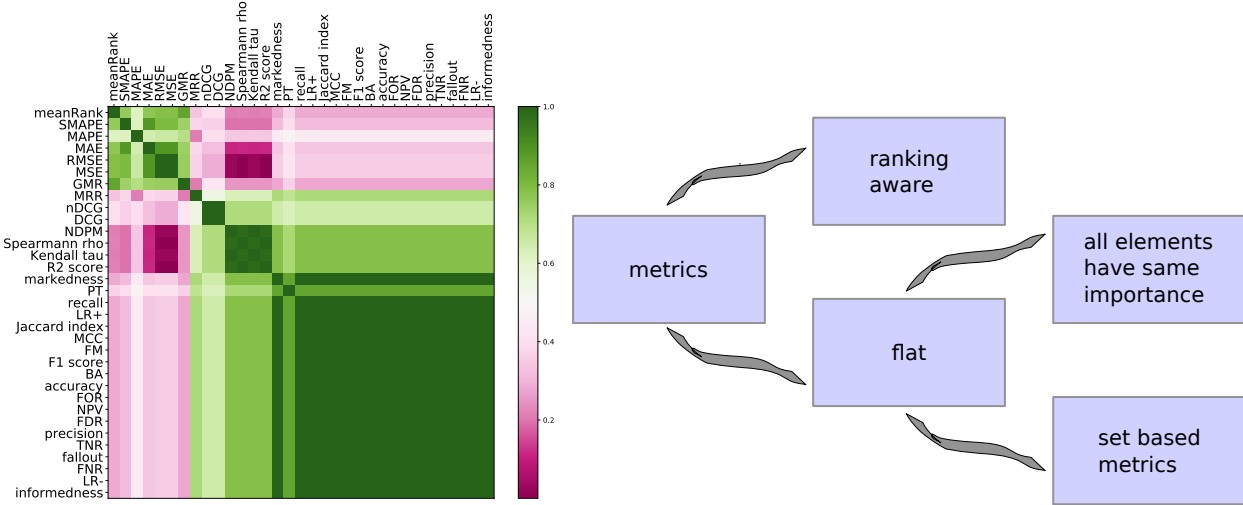

Figure 1: **On the left:** Heatmap of the agreement ratios among pairs of ranking evaluation metrics. **On the right:** The theoretical subdivision of the metrics; the

Recommender Systems. We refer to the group as ranking evaluation metrics; a complete list is summarized in Table 1.

We categorize the ranking evaluation metrics under two different theoretical aspects. One subdivision derives from their "awareness" of the position of single items in the rankings: *Ranking aware metrics* satisfy this criterion while *flat metrics* do not. In this second group, we find the *set-based metrics* and the ones assigning equal importance to each position. The subdivision is shown in Figure 1 on the right. From their theoretical definition, we individuate four main groups: *confusion matrix-based CMB metrics* focus on the number of correctly retrieved elements and are essentially set-based metrics; *correlation-based CB metrics* quantify the ordinal association between the two rankings from a statistical perspective; *error-based EB metrics* are often used to analyze the performance of predicting models and are flat metrics assigning equal importance to each position; Finally, *cumulative gain-based CGB metrics* focus on the rankings of the single elements; additional explanations on how the single metrics have been classified and the definition of the single metrics are available in Appendix B.

## 4 Ranking evaluation metrics on symmetric groups

To generalize the metrics over an abstract structure, we introduce *symmetric groups* $S_n$. Given a finite set $\mathcal{N} = \{1, \ldots, n\}$, the *symmetric group* $S_n$ is the set of bijective functions from $\mathcal{N}$ to $\mathcal{N}$, i.e., the rankings or *permutations* of elements in $\mathcal{N}$; $S_n$ has size $n!$. Permutations are designed with lowercase Greek letters, i.e., $\sigma \in S_n$. Exceptionally, id indicates the group identity or *identity function*; the *identity function* defines

mathematically the supposition that the identical ordering assigns to each 'item' $i$ its position $i$, i.e., that the items' names correspond to their positioning. $\sigma(i)$ indicates the position in which item $i$ is sent by $\sigma$ and, given $\sigma, \nu \in S_n$, $\sigma \circ \nu \in S_n$ is a new ranking defined by $\sigma \circ \nu(i) = \sigma(\nu(i)), \forall i \in \{1, \ldots, n\}$; $\circ$ is the group operation and it is not commutative, i.e., generally $\sigma \circ \nu \neq \nu \circ \sigma$. $\sigma_{|k} = (\sigma(1), \ldots, \sigma(k))$ indicates the ranking of the first $k$ elements; metrics@$k$ consider exclusively the first $k$ ranked elements. Finally, a *(single) swap* is a permutation $\sigma = (j \ k) \in S_n$, swapping only the two elements $j, k$ in $\mathcal{N}$; Hassanzadeh & Milenkovic (2014) refers to them as *transpositions*.

## 4.1 Clustering by agreement

Our work is mainly justified by the lack of "consistent" evaluation of rankings when using different metrics. A *ranking evaluation metric* is a function $m : S_n \times S_n \to \mathbb{R}_+$, taking two permutations as input and returning a real number. In some cases, metrics take only one ranking as input; we refer to them as *single input metrics*. All the given definitions work correspondingly for one-input metrics.

**Definition 1.** *Two metrics $m_1, m_2$ are* non-consistent *if there exists $\sigma, \mu, \nu \in S_n$ such that the following two conditions hold:*

$$
\begin{aligned}
m_1(id, \sigma) \leq m_1(id, \mu) \ &\wedge \ m_2(id, \sigma) \leq m_2(id, \mu) \\
m_1(id, \sigma) \leq m_1(id, \nu) \ &\wedge \ m_2(id, \sigma) > m_2(id, \nu)
\end{aligned}
\tag{1}
$$

*Otherwise, we say that $m_1, m_2$ are* consistent.

The first line of equation 1 guarantees that the reversed metric $\tilde{m}_2 = -m_2$ is still non-consistent with $m_1$. Proving consistency between two metrics is much trickier than finding three rankings satisfying the inconsistency condition; therefore, rather than classify them, we estimate the degree of inconsistency among pairs of metrics by introducing the *agreement ratio*. The coefficient provides an estimate of the extent to which two metrics disagree in the evaluation of rankings over symmetric groups.

**Definition 2.** *For any $\sigma \in S_n$ fixed, the $\sigma$* agreement ratio *among two ranking evaluation metrics, $m_1$ and $m_2$ is*

$$
AR^{\sigma}_{m_1, m_2} = \frac{1}{|\mathcal{P}(S_n)|(|\mathcal{P}(S_n)| - 1)} \sum_{\mu, \nu \in \mathcal{P}(S_n), \mu \neq \nu} f^{m_1, m_2}_{\sigma}(\nu, \mu)
$$

*where $f^{m_1, m_2}_{\sigma}(\nu, \mu) = \mathbb{1}\{m_1, m_2$ are consistent w.r.t. $\sigma$ on the rankings $\mu, \nu\}$ (or equivalently $f^{m_1, m_2}_{\sigma}(\nu, \mu) = \mathbb{1}\{$equation 1 is not satisfied$\}$), $\mathbb{1}$ is the indicator function and $\mathcal{P}(S_n)$ is the power set over $S_n$.*

As the size of $\mathcal{P}(S_n)$ grows exponentially, we randomly sample a subset $\mathcal{T}$ of $\mathcal{P}(S_n)$ thus obtaining an estimate of the number of inconsistencies existing among two metrics. The agreement ratio equals 1 if $m_1$ and $m_2$ are consistent and goes to zero with increasing inconsistencies found; furthermore, the agreement ratio is a symmetric metric.

The color-code heatmap in Figure 1 highlights, respectively, in green and pink, the existence of a high agreement and disagreement; a partial agreement is represented in white. It is visible that similarly theoretically defined metrics as grouped Table 1 tend to have an agreement ratio closer to 1. The agreement ratio represents an estimate of the number of inconsistencies among metrics; Figure 1 refers to rankings in $S_{100}$, where $\mathcal{T}$ contains 10000 random rankings. For CMB metrics, we fixed to 30 the number of retrieved and relevant elements. We use as reference ranking $\sigma$ the identity function previously defined; however, we would get similar colored heatmaps using other reference rankings.

## 5 Properties for ranking evaluation metrics

Most pairs of metrics are affected by frequent inconsistent evaluations (cf. Section 4.1). We list essential mathematical properties to highlight the peculiarity of each metric and give the chance to properly select one or another based on them for a context-dependent evaluation. The properties in question are: (1) *identity of indiscernibles* (IoI); (2) *symmetry* (or *independence from a ground truth*); (3) *robustness* (Type-I and Type-II); (4) *stability* with respect to $k$; (5) *sensitivity* and *width-swap-dependency*; (6) (induced) *distance*. Some of

| ranking length | relevant | baseline | $\sigma$ | $\tau$ | CMB metrics | MSE | RMSE | MAE | MAPE | SMAPE | $R^2$ score | Kendall's $\tau$ | Spearmann $\rho$ | DCG | nDCG | MRR | GMR | NDPM | meanRank |
|---|---|---|---|---|---|---|---|---|---|---|---|---|---|---|---|---|---|---|---|
| 10 | 5 | id | (1 2) | id | ○ | ● | ● | ● | ● | ● | ● | ● | ● | ● | ● | ● | ● | ● | ● |
| 10 | 5 | id | (1 2) | (3 4) | ○ | ○ | ○ | ○ | ● | ● | ○ | ○ | ○ | ● | ● | ○ | ○ | ○ | ○ |
| 10 | 5 | id | (1 2) | (2 4) | ○ | ● | ● | ● | ○ | ○ | ● | ● | ● | ● | ● | ○ | ○ | ● | ○ |

Table 2: Examples of rankings that metrics cannot distinguish. We compare for each evaluation metric $m$ the values $m(\mathrm{id}, \sigma)$ and $m(\mathrm{id}, \tau)$. If the metric fails in distinguishing the two rankings, we impute a ○; else, a ●.

them have been defined in other domains, wee.g., Gösgens et al. (2021b;a) define the symmetry property for cluster similarity indices and metrics for classification models, Hassanzadeh & Milenkovic (2014) defines the "resistance to relabeling" in the context of rank aggregation, Cook et al. (1986); Fligner & Verducci (1986) define the importance of constructing distances for partial orderings; will refer to each of them in the respective sections.

For each property, we will highlight in which context and why it is important. Table 3 and Table 4 help the reader to keep trace of the mentioned results. The code will be on GitHub upon acceptance [1].

## 5.1 Identity of indiscernibles

Ideally, a metric $m$ quantifies how "close" or "similar" two rankings $\sigma$ and $\tau$ are. However, situations may arise where $\sigma$ and $\tau$ are "so" similar to be practically indistinguishable by some metrics. This effect might be undesired in some fields, such as (fair) rank aggregation, where even small differences, especially in the presence of protected groups, make the difference between fair and unfair rankings.

**Definition 3.** *A metric $m$ satisfies the* identity of indiscernible (IoI) property *if, $\forall \sigma \in S_n$ fixed, the following holds*

$$m(\sigma, \tau) = m(\sigma, \nu) \Leftrightarrow \tau = \nu, \qquad \forall \tau, \nu \in S_n. \tag{2}$$

Up to renaming the elements, we can rewrite Equation equation 2 as $m(\mathrm{id}, \tau) = m(\mathrm{id}, \nu) \Leftrightarrow \nu = \tau$ where id is the usual identity of $S_n$.

Almost all metrics do not satisfy the IoI property; examples are set-based metrics and metrics$_{@k}$, i.e., where a metric $m_{@k}$ evaluates only the top $k$ elements of the rankings . Clear examples not satisfying equation 2 are rankings $\sigma$ that can be written as a disjoint composition of cycles of permutations of elements before and after $k$ Hall (2018). Table 2 illustrates examples for each metric where the IoI is not satisfied. It can be proven that

**Proposition 5.1.** *DCG and nDCG satisfy the IoI property.*

The proof finds place in Appendix A.

## 5.2 Symmetry property

Often, guarantees that the evaluation is symmetric with respect to input items are desirable Gösgens et al. (2021a;b), particularly when the interest is in having a sort of mathematical distance, e.g., for rank aggregation. However, as usual, the context rules the need for a symmetric evaluation. The symmetry property studies whether the metric's evaluation is independent of the order in which the rankings are compared. In RS and IR, the common presence of a "ground truth order" makes the symmetric property impossible.

---

[1] https://anonymous.4open.science/r/rankingsmetrics/README.md

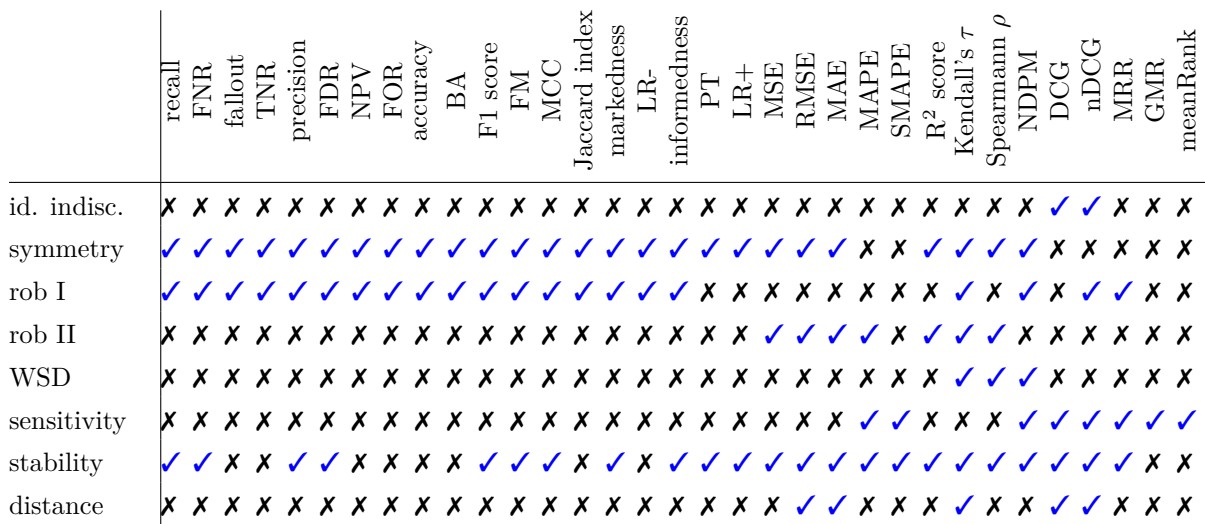

| | recall | FNR | fallout | TNR | precision | FDR | NPV | FOR | accuracy | BA | F1 score | FM | MCC | Jaccard index | markedness | LR- | informedness | PT | LR+ | MSE | RMSE | MAE | MAPE | SMAPE | R² score | Kendall's τ | Spearmann ρ | NDPM | DCG | nDCG | MRR | GMR | meanRank |
|---|---|---|---|---|---|---|---|---|---|---|---|---|---|---|---|---|---|---|---|---|---|---|---|---|---|---|---|---|---|---|---|---|---|
| id. indisc. | ✗ | ✗ | ✗ | ✗ | ✗ | ✗ | ✗ | ✗ | ✗ | ✗ | ✗ | ✗ | ✗ | ✗ | ✗ | ✗ | ✗ | ✗ | ✗ | ✗ | ✗ | ✗ | ✗ | ✗ | ✗ | ✗ | ✓ | ✓ | ✗ | ✗ | ✗ | ✗ | ✗ |
| symmetry | ✓ | ✓ | ✓ | ✓ | ✓ | ✓ | ✓ | ✓ | ✓ | ✓ | ✓ | ✓ | ✓ | ✓ | ✓ | ✓ | ✓ | ✓ | ✓ | ✓ | ✓ | ✓ | ✓ | ✓ | ✗ | ✗ | ✓ | ✓ | ✓ | ✓ | ✗ | ✗ | ✗ |
| rob I | ✓ | ✓ | ✓ | ✓ | ✓ | ✓ | ✓ | ✓ | ✓ | ✓ | ✓ | ✓ | ✓ | ✓ | ✓ | ✓ | ✓ | ✓ | ✓ | ✗ | ✗ | ✗ | ✗ | ✗ | ✗ | ✓ | ✗ | ✓ | ✗ | ✓ | ✓ | ✗ | ✗ |
| rob II | ✗ | ✗ | ✗ | ✗ | ✗ | ✗ | ✗ | ✗ | ✗ | ✗ | ✗ | ✗ | ✗ | ✗ | ✗ | ✗ | ✗ | ✗ | ✗ | ✓ | ✓ | ✓ | ✓ | ✗ | ✓ | ✓ | ✓ | ✗ | ✗ | ✗ | ✗ | ✗ | ✗ |
| WSD | ✗ | ✗ | ✗ | ✗ | ✗ | ✗ | ✗ | ✗ | ✗ | ✗ | ✗ | ✗ | ✗ | ✗ | ✗ | ✗ | ✗ | ✗ | ✗ | ✗ | ✗ | ✗ | ✗ | ✗ | ✗ | ✓ | ✓ | ✓ | ✗ | ✗ | ✗ | ✗ | ✗ |
| sensitivity | ✗ | ✗ | ✗ | ✗ | ✗ | ✗ | ✗ | ✗ | ✗ | ✗ | ✗ | ✗ | ✗ | ✗ | ✗ | ✗ | ✗ | ✗ | ✗ | ✗ | ✗ | ✗ | ✓ | ✓ | ✗ | ✗ | ✗ | ✓ | ✓ | ✓ | ✓ | ✓ | ✓ |
| stability | ✓ | ✓ | ✗ | ✗ | ✓ | ✓ | ✗ | ✗ | ✗ | ✗ | ✓ | ✓ | ✓ | ✗ | ✓ | ✗ | ✓ | ✓ | ✓ | ✓ | ✓ | ✓ | ✓ | ✓ | ✓ | ✓ | ✓ | ✓ | ✓ | ✓ | ✓ | ✗ | ✗ |
| distance | ✗ | ✗ | ✗ | ✗ | ✗ | ✗ | ✗ | ✗ | ✗ | ✗ | ✗ | ✗ | ✗ | ✗ | ✗ | ✗ | ✗ | ✗ | ✗ | ✓ | ✓ | ✗ | ✗ | ✗ | ✓ | ✗ | ✗ | ✓ | ✓ | ✗ | ✗ | ✗ | ✗ |

Table 3: Summary table of the property satisfied by the metrics.

**Definition 4.** *A metric* $m : S_n \times S_n \to \mathbb{R}$ *is* symmetric *if*

$$m(\sigma, \nu) = m(\nu, \sigma), \qquad \forall \sigma, \nu \in S_n. \tag{3}$$

### 5.3 Robustness

The IoI property studies whether metrics can distinguish rankings, regardless of their similarity. On the other side, the similarities among rankings should be projected on the evaluations: Small differences in rankings should result in small differences in the evaluation scores. Under the assumption that a single swap represents a small difference between two rankings, the *Type I robustness* property assesses how sensitive a ranking evaluation metric is to single swaps in the compared rankings.

**Definition 5.** *A metric m is* Type I Robust *if a* single swap *in one of the rankings implies small changes in its evaluation, i.e.,*

$$|m(\sigma, \nu) - m(\sigma, \nu \circ (i\ j))| < \epsilon, \qquad \forall \sigma, \nu \in S_n. \tag{4}$$

We compute the average of the results of equation 4 evaluated on a set $\mathcal{I} \subseteq S_n \times S_n$ of 1000 different randomly drawn pairs of rankings in $S_{100}$, i.e., $\sum_{(\sigma,\nu)\in\mathcal{I},(i,j)\in\{1,\cdots,n\}^2} |m(\sigma, \nu) - m(\sigma, \nu \circ (i\ j))|$ and round it to two decimal numbers. We state that the metric satisfies the Type I Robustness if the resulting average is 0.

For completeness, we define a second type of robustness that studies the effect of renaming the items in the rankings. Diaconis (1988) mentions Type II Robustness as "right-invariance" and Hassanzadeh & Milenkovic (2014) as "resistance to item relabeling".

**Definition 6.** *A metric is* Type II Robust *if it is an invariant w.r.t. the composition of permutations, i.e., it holds*

$$m(\mu, \sigma) = m(\mu \circ \nu, \sigma \circ \nu), \forall \mu, \sigma, \nu \in S_n.$$

*Type II Robustness* property investigates whether applying the same change in both rankings affects the evaluation. The property is essential in contexts where the numbers appearing in the rankings have to be considered as proper "items" or "items' names"; this is often the case in rank aggregation approaches, Recommender Systems, and Information Retrieval techniques. However, it does not apply when dealing with importance scores. We claim that

**Proposition 5.2.** MSE, RMSE, MAE, MAPE, $R^2$ score, Kendall's $\tau$ score *and* Spearmann's $\rho$ *are the only considered metrics satisfying the Type II Robustness.*

The proof derives directly from their definitions (see Appendix A).

### 5.4 Sensitivity

The sensitivity property is valuable for a metric, particularly in the case of Recommender Systems and Information Retrieval, where high dimensional rankings may not be fully explored. Under the assumption that a full exploration of the rankings is not possible, sensitive metrics assign more weight to the first part of the rankings, considering whether the first $k$ items are "correctly" ranked. Mathematically, we define the width of a swap $(i\ j) \in S_n$ being the quantity $|i - j|$.

**Definition 7.** *Given $i < j < k < l \in \{1, \ldots, n\}$ and $(i\ j), (l\ k)$ having the same width. A ranking evaluation metric $m$ is* sensitive *if $\exists \sigma \in S_n$ such that it holds $m((i\ j) \circ \sigma) \neq m((k\ l) \circ \sigma)$.*

As the evaluation of the property is far from easy, we introduce the *width swap dependency*, formalizing a property that prevents the metrics from being sensitive.

**Definition 8.** *Given a swap $(i\ j) \in S_n$ and $|i - j|$ its* width*, $m$ is* width swap dependent *(WSD) if it evaluates swaps with the same width equally, i.e., $m((i\ j)) = m((k\ l))$ if $|i - j| = |k - l|$ holds; otherwise, it is called* non-width swap dependent.

The WSD property cuts out some of the metrics from being sensitive. From their definitions, it can be proven that

**Lemma 5.3.** *Kendall's $\tau$, Spearmann $\rho$, NDPM are width swap independent.*

The proof finds place in Appendix A. For the other metrics, it is trivial to find pairs of disjoint swaps had different effects in the final evaluation when happening at various positions within the rankings.

### 5.5 Stability

We introduce the stability property for those metrics that can be applied on "rankings @$k$". We recall that a ranking at $k$ is the ranking of the items in the first $k$ positions. To evaluate rankings @$k$, it is essential that the difference between evaluations "@$k$" and "@$k+1$" is not significant, i.e., that the choice of $k$ does not highly impact the result; this guarantees a trustworthy evaluation.

**Definition 9.** *A ranking evaluation metric $m$ is* stable *if, for any two rankings $\sigma, \nu \in S_n$, it holds*

$$|m_{@k-1}(\sigma, \nu) - m_{@k}(\sigma, \nu)| < \epsilon_k \tag{5}$$

*with $\epsilon_k$ small. Moreover, the sequence $\{\epsilon_k\}_k$ satisfies $\lim_{k \to n} \epsilon_k = 0$.*

The property is again essential for extremely long rankings and for contexts where rankings are not fully explored. We evaluate the stability by randomly drawing 1000 pairs of rankings in $S_{100}$, computing the absolute differences of equation 5, and counting the number of times that equation 5 holds with $\epsilon_k = \frac{1}{k}$. We state that a metric is stable if the criterion is satisfied in at least 97.5% of the cases.

### 5.6 Distance

In mathematics, the terms metric and distance are synonyms. However, when it comes to evaluation metrics, most of them are not "distances" on $S_n$ in the mathematical sense. Whether a metric is a mathematical distance or not is often insignificant for the final evaluations; however, being aware of this fundamental mathematical difference can avoid incomprehension and misuses.

**Definition 10.** *A* distance *on a set $X$ is a function $f_m : X \times X \to [0, \infty) : (x, y) \mapsto f_m(x, y) \in \mathbb{R}_+$ that, for all $x, y, z \in X$, satisfies:*

1. *$f_m(x, y) = 0 \Leftrightarrow x = y$,*

2. *the* positive definiteness*, i.e., $f_m(\sigma, \nu) \geq 0, \forall \sigma, \nu \in X$,*

3. *the* symmetry *property and*

    *4. the* triangle inequality, *i.e.,* $f_m(x,y) \leq f_m(x,z) + f_m(z,y)$.

Some ranking evaluation metrics are distances; In Hassanzadeh & Milenkovic (2014); Diaconis (1988), it is proven that Kendall's $\tau$ is a distance. However, a ranking evaluation metric that does not satisfy some of the properties mentioned in Definition 10 is not a distance.

We investigate if we can induce distances from single input metrics. Given a metric $m : S_n \to \mathbb{R}$, we consider two options as potential induced distances, i.e., $f_m(\sigma, \nu) = m(\sigma) - m(\nu)$ or $\tilde{f}_m(\sigma, \nu) = |m(\sigma) - m(\nu)|$. DCG and nDCG are the only two metrics satisfying the IoI property that, for metrics with one unique argument, is equivalent to Property (1) for $f_m$. We can easily prove that

**Proposition 5.4.** $f_m$ *is* not *a distance while* $\tilde{f}_m$ *is a distance with the IoI property, where $m$ is either DCG or nDCG.*

The formal proof can be found in Appendix A.

## 6   Are the metrics interpretable? Thoughts over maximal and minimal agreement properties

Given the importance of trust, fairness, and explainability for machine learning methods, one could then ask how "interpretable" the scores assigned by the metrics are. We first need some definitions.

**Definition 11.** *A ranking evaluation metric $m$ is said to satisfy the* maximal agreement property *if (a)* $m(\sigma, \sigma) = m_{\max}, \forall \sigma \in S_n$ *and (b)* $m(\sigma, \nu) \leq m_{\max}, \forall \nu, \sigma \in S_n$. *We say that $m$ is* lower-bounded *if it exists a real number $m_{\min}$ such that $m(\sigma, \nu) \geq m_{\min}, \forall \nu, \sigma \in S_n$. An evaluation metric that admits a lower bound is said to satisfy the* minimal agreement *property.*

For a metric to be "interpretable" we expect that

1. each ranking is maximally similar to itself and, given $n \in \mathbb{N}$, this value is constant, i.e., $m(\sigma, \sigma) = m_{\max}, \forall \sigma \in S_n$ and $\forall n$;

2. $m$ satisfies the maximal agreement property;

3. there exists a lower bound $m_{\min}$ for any possible pair of rankings, i.e., $m(\sigma, \mu) \geq m_{\min}, \forall \sigma, \mu \in S_n$.

Exemplary is the Kendall's $\tau$ metric which satisfies $m(\sigma, \sigma) = 1$ and $m(\sigma, \mu) \in [-1, 1]$ for all $\sigma, \mu \in S_n$. The maximal agreement property says that each ranking is maximally similar to itself, and no other ranking can achieve a higher score than $m_{\max}$; furthermore, ideally, $m_{\max}$ is independent of the length of the rankings. Properties (1) and (2) imply that a ranking evaluation metric is a monotone increasing function of the similarity of two rankings: the more similar two rankings are, the higher the score they get when evaluated using an "interpretable" metric. Having that $m_{\max}$ is independent of $n$ is a necessary condition for having an evaluation of rankings independent of $n$. However, this is hardly satisfied by any metrics, and only after introducing a normalization score do the metrics satisfy the requirement. Furthermore, the lowest scores are assigned by some metrics to maximally similar pairs of rankings, e.g., error-based metrics. The only metrics, among the ones considered in this paper, automatically satisfying this property are Kendall's $\tau$ score and Spearmann $\rho$.

A ranking evaluation metric satisfying the maximal agreement property is also *upper-bounded*. For the sake of interpretability, we could check whether a metric $m$ satisfies $m(\rho^{-1}, \rho) = m_{\min}$ where $\rho^{-1}$ indicates the inverse ranking. How do we define the inverse of a ranking? Kendall's $\tau$ satisfies this property, given that the inverse of one ranking $\sigma$ is the ranking $\tau$ assigning the highest position to the last element of the ranking $\sigma$; however, this does not correspond with the inverse of the ranking in the symmetric group. Assessing whether metrics for permutations are humanly interpretable is not new and has already been discussed in Diaconis (1988). However, then, as well as now, the concept of interpretability lacks a unified definition. Thus, we leave this section open and do not argue further on the interpretability of the considered metrics.

| | description | domain |
|---|---|---|
| identity of indiscernibles | in highly sensitive evaluations, where detecting tiny differences among rankings is essential | (Fair) rank aggregation
Recommender Systems
Feature ranking/selection |
| symmetry | ensures that the input rankings have an equal role in the evaluations | Rank aggregation
Contexts independent from ground truths |
| robustness I | ensures that small changes influence proportionately the evaluations | Information Retrieval
Rank aggregation |
| robustness II | ensures independence from items renaming | Information Retrieval
Rank aggregation
Feature ranking
Rank aggregation |
| sensitivity | for not fully explored rankings, when the interest is on to the top part of the rankings | Information Retrieval
Recommender Systems |
| stability | ensures trustworthiness in evaluations @$k$ | Information Retrieval
Recommender Systems |
| distance | ensures that the metric in questions respect the definition of distance on $S_n$ | (Fair) rank aggregation |

Table 4: Summary of the properties

## 7 Discussion

We explored metrics for comparing and evaluating rankings and analyzed their theoretical properties. All the mentioned metrics are widely used in the literature to evaluate Recommender Systems, Information Retrieval, feature ranking, rank aggregation methods, and items' score assignments. Each property is highly desirable in some contexts and less in others. The IoI property is desirable in highly sensitive evaluations, where detecting tiny differences among rankings is essential; fair ranking aggregation is an example, where swapping items can make the difference between fair and unfair rankings. Conversely, robustness ensures that small changes influence the evaluations proportionately in a one-to-one fashion. A metric that satisfies both the IoI and the robustness properties ensures contemporaneously that small changes are not overlooked but do not significantly impact the evaluations. The symmetry property ensures that the input rankings have an equal role in the evaluations. This is essential in most domains unless ground truth ranking is available. Note that non-symmetric metrics are also not distances. Rank aggregation is again an example of use for the symmetry property, where the consensus ranking is directly compared with the original rankings provided. Sensitivity is crucial when rankings are not fully explored. This is often the case for Recommender Systems and Information Retrieval techniques' evaluations. With the same applicability, the stability property ensures trustworthiness in evaluations @$k$, which is again highly relevant for Recommender Systems and Information Retrieval techniques. To assure stable evaluations, we recommend considering evaluating the impact @$k$ and @$(k+i)$ with $i$ arbitrarily chosen, in particular when $k << n$. Finally, the distance property is defined to complete the proposed analysis and highlights the chance that mathematical terms are misused in machine learning contexts. Table 4 summarizes the properties' descriptions and application domains.

## 8 Conclusion

Throughout the paper, we explored metrics widely used in the literature to evaluate Recommender Systems, Information Retrieval, feature selection, and rank aggregation methods; rankings are the common output

of all these methods. We observed a common presence of non-consistent evaluations of rankings, deriving from the different definitions of the ranking evaluation metrics. Focusing on a mathematical perspective and viewing rankings as elements of symmetric groups and the metrics as functions defined over mathematical groups, we list a set of well-founded mathematical properties for ranking evaluation metrics. The differences among metrics are highlighted by the differences in the satisfiability of the properties, thus grounding the reasons for the inconsistencies in the evaluations. As each property is highly desirable in some contexts and less in others, we summarize the obtained insights that can be of immediate use when looking for an appropriate metric for a specific domain.

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
