# OpenReview forum: "Ranking evaluation metrics from a group-theoretic perspective"
_TMLR — Rejected by TMLR_

### Review · Reviewer_QFtL · 2024-05-20

**Summary Of Contributions:**

The authors present a comparison of evaluation metrics for ranking problems, from a theoretical perspective of group theory. The authors started by categorizing all ranking metrics into several categories based on the construction of the metrics. The authors then describe several desirable theoretical properties of ranking metrics, followed by analyses of which metrics satisfy the properties. Finally, the authors discuss several domain applications, and which theoretical properties are relevant to the applications.

**Audience:**

Yes

**Broader Impact Concerns:**

I do not have any broader impact concerns for this paper.

**Claims And Evidence:**

No

**Requested Changes:**

Please address the concern I have in the weakness section above. For the clarity issue, I recommend the authors to use a professional editor service, to further improve the clarity of the paper.

**Strengths And Weaknesses:**

**Strengths**:

- Interesting work on comparing various metrics for ranking problems.
- It would be likely to help practitioners in deciding what metrics should they use in their projects.
- The properties of metrics postulated in the paper are well-motivated and correlate with some practical use cases.
- The authors explain each property in detail and provide analyses on which metrics satisfy the property.
- The authors provide a discussion at the end on how those properties correlate with practical applications in several domain areas.

**Weaknesses/Questions**:

- One of the main weaknesses of the paper is in the clarity. On many occasions, the sentences in the paper are hard to parse. This can be due to very long sentences that consist of multiple ideas, word choices that are less suitable, a few problems in the grammar, or unclear utilization of mathematical symbols. This issue makes the paper harder to understand.
- The organization of the content could also be improved. The current format missed the definition of each performance metric in the main paper, which, I think is one of the most important parts of the content, as the paper studies the properties of each metric.
- Some of the terminologies are not clearly explained, for example, what does "id" refer to? The authors only describe it as the "identity function", but do not explain the meaning in the context of ranking.
- In some of the properties, the authors only explain the properties of some metrics and leave out the rest.
- In Table 4, the authors provide a discussion on what domains/areas the metric properties are considered to be important. It would be good if the author could explain in more detail on why the metric is important, and fully equip them with references to previous studies to back up the claim.
- Another related table that is important for practitioners that do the reverse mapping from Table 4 is also good to have. What I meant is the mapping from a domain/area, then what properties are important to the domain, followed by the suggested metrics for the domain that satisfy the desired properties.
- Novelty and relation to the previous work. It would be good if the authors could describe more about which parts of the paper are new, and which parts that have already been proposed by previous studies. Are all of the properties discussed in the paper new contributions, or some of them have been studied by previous works?

---

### Review · Reviewer_4oAK · 2024-06-03

**Summary Of Contributions:**

This submission reviews evaluation metrics for rankings, viewing rankings as elements of the symmetric group $S_n$. First, a categorization of metrics into groups is proposed. Next, the rate of agreement between metrics is studied empirically. The main part of the submission proposes desirable mathematical properties for ranking evaluation metrics and shows which properties are satisfied by which metrics.

**Audience:**

Yes

**Claims And Evidence:**

No

**Requested Changes:**

Addressing all of the weaknesses above is critical for me. Addressing the minor comments should be straightforward.

**Strengths And Weaknesses:**

### Strengths
- The submission has the potential to be a useful survey and categorization of ranking evaluation metrics.
- I think the elucidation of which properties are satisfied by which metrics has value.

### Weaknesses
Overall, I think the presentation is still incomplete and unclear.
1. I got very little out of Section 3 in its current form. Below I mention two specific aspects. Only later did I find that Appendix B presents the four groups of evaluation metrics and also defines (albeit in compressed form) the metrics listed in Table 1. However, I did not see a reference to Appendix B in the main text. In any case, I think that Appendix B needs to be brought into the main text. Otherwise, the paper is not accessible to readers who are not already experts in ranking evaluation metrics.
    - I do not understand the subdivisions shown on the right side of Figure 1 and described informally in the text. I think these subdivisions should be defined mathematically.
    - Similarly, the listing of the four groups of metrics does not mean much to me without first introducing some mathematical formalism and then describing the groups either mathematically or with more detailed text.
1. Unclear definitions (looking back, it appears that I have questions about almost all the definitions):
    1. Definition 2:
        - "$\mu,\nu$ are consistent w.r.t. $\sigma$": Does this mean that (1) does not hold for the given triplet $\mu, \nu, \sigma$ and metrics $m_1, m_2$? Definition 1 defines consistency of metrics $m_1, m_2$ but not of rankings $\mu, \nu$.
        - The text mentions that $\mu, \nu$ are randomly sampled. What about $\sigma$? Is it also sampled and averaged over to obtain a $\sigma$-independent agreement ratio (the left side of Figure 1 does not mention a specific $\sigma$)?
    1. Definition 3: I am confused by the phrase "$\forall\sigma \in S_n$ fixed." Does $m(\sigma, \tau) = m(\sigma, \nu)$ have to hold for all $\sigma$ to imply $\tau = \nu$? Or only for some $\sigma$?
    1. Definition 4: I do not understand why the "presence of a 'ground truth order' makes the symmetric property impossible." Couldn't we have a symmetric metric where one of the input rankings happens to be the ground truth?
    1. Definition 5:
        - Should $\epsilon$ be a parameter of the definition, since it is not specified?
        - Does (4) have to hold for all $\sigma, \nu$?
        - What does "average of the results of equation (4)" mean? Is the "result" an indicator that equals 1 if (4) is violated? Does this sentence refer to a figure that is missing?
    1. Definition 6: Should this be for all $\mu, \sigma, \nu$?
    1. Definition 7: What is the definition of "impact" here?
    1. Definition 8: I think the authors mean that the metric depends only on the width of the swap. If so, I recommend finding better terms than "(non-)width swap dependent," perhaps using the word "invariant" instead.
    1. Definition 9: How is $m_{@k}$ defined?
    1. Above Proposition 5.4: The paper does not define "single input metrics."
    1. Definition 11:
        - The phrase "given $n \in \mathbb{N}$, this value is constant" suggests that $m_{\max}$ could depend on $n$, but later, it is stated that "ideally, $m_{\max}$ is independent of the length of the rankings." Which is it?
        - Does property 2 (maximal agreement) imply property 1?
1. I think the manuscript (still) does not do enough in discussing differences with related work. Of particular note:
    1. I think the work of Diaconis (1988) should be discussed in greater detail as it sounds the most similar. The text mentions that "some of these properties [referring to Diaconis] are similar to the ones we define." Which ones? What are the differences? Which metrics does Diaconis (1988) focus on?
    1. Section 5, first paragraph, "some of [these properties] have been defined in other domains": Which properties and in which prior works? Are there any differences?
1. Unjustified statements in Section 6
    1. "Properties (1) and (2) imply that a ranking evaluation metric is a monotone increasing function of the similarity of two rankings": The "similarity of two rankings" has not been defined, so I do not know how this statement can be assessed.
    1. "If $m_{\max}$ is independent of $n$, the evaluation of rankings is independent of $n$": No, the maximum value being independent of $n$ does not imply that other evaluations of the metric are independent of $n$.

1. Below Section 5.3, "for the other metrics, we evaluate if pairs of disjoint swaps had different effects": Where is this done?

### Minor comments
1. Figure 1 caption: Should "disagreement ratios" be "agreement"?
2. Definition 2: Is the sum only over distinct pairs $\mu \neq \nu$?
3. Section 4, last paragraph: What is meant by "theoretical similar metrics"?
4. Above Proposition 5.1: What does "disjoint composition of cycles of permutations" mean?
5. Below Definition 6, "the numbers appearing in the rankings have to be considered as proper 'items' or 'item's names'": Does "numbers appearing in the rankings" refer to the labels $1, \dots, n$ of the set $\mathcal{N}$?
6. Perhaps Section 6 should have a more modest title related to the maximal/minimal agreement property that is actually discussed.

---

### Review · Reviewer_h2Jp · 2024-06-19

**Summary Of Contributions:**

This paper considers metrics that are used to evaluate models which output rankings, such as in the case of recommender systems or information retrieval. The authors analyze ranking metrics from the perspective of symmetric groups on sets of finite elements. They define 8 properties which can be used to describe ranking metrics, and they indicate for a range of popular metrics whether or not they satisfy each of the 8 properties.

As a caveat to my review, I am not an expert in these metrics or in areas where they are commonly used, such as recommender systems or information retrieval. Therefore, it is difficult for me to say with any confidence how novel these analyses are.

**Audience:**

Yes

**Claims And Evidence:**

No

**Requested Changes:**

The weaknesses above all need to be addressed in some form. The most important things are: making sure definitions are precise and complete, clarifying the scope of the problem both in a mathematical sense and in terms of real world applications, and providing some concrete examples and/or richer discussion about how this framework could be applied in practice.

**Strengths And Weaknesses:**

Strengths
---------
The goal of providing a unifying analysis of many different metrics is clearly a useful one. The mathematical framework they apply is clear and straightforward, and it seems comprehensive in the sense that it could be used to derive many additional properties beside the 8 the authors identified. The 8 properties they derive seem useful on the whole for comparing different ranking metrics. The authors acknowledge that different metrics may be useful in different domains.


Weaknesses
----------
While unifying frameworks are good for generating insights, I find it unclear how the authors envision this framework being used and what problems they think it will address. They allude to potential "misuses" of ranking metrics, but they don't really provide any examples, other than noting that in fair ranking tasks, small differences in rankings can mean the difference between fair and unfair rankings. It's fairly self-evident that the evaluation of model performance depends on the metric used, and that different metrics may yield different conclusions about how two models compare. So while it's helpful to formalize some of the differences between metrics, it's not clear what a downstream user might do with this information.

The authors seem to aim to address this: they say "we look to answer the question 'which mathematical properties are essential for evaluating rankings?'" However, I don't think the paper does answer this in a satisfactory way. There is minimal or no discussion of why these 8 properties were chosen rather than other possible properties, when these properties are more or less desirable, or how a failure to take these properties into consideration could cause someone to misuse a metric somehow. It would be extremely helpful to have some concrete examples. (Even more compelling would be an empirical analysis of how widespread such "misuses" are, though this is certainly not essential to the paper.) It's just not clear to begin with what constitutes misuse of a metric, so it's hard for the reader to fill in the gaps.

The scope of the problem is also unclear in the sense that the authors include in their analysis metrics like MAE and MSE, which they describe as "assigning equal importance to each position" in a ranking. Given that their mathematical framework considers permutations of integers $\{1, \ldots, n\}$, I'm confused what these metrics refer to. Are they implicitly also considering metrics that are used to evaluate lists of real numbers, like relevance scores? But then this doesn't fit into their mathematical framework. If these metrics are envisioned as operating on the indices $\{1, \ldots, n\}$ then they can't be understood as "assigning equal importance to each position." I don't know how to reconcile this.

I'm not sure what the value of the section on distances (section 5.6) is. It's true that many evaluation metrics are not distances in the mathematical sense; it's also true, as the authors point out, that ``Whether a metric is a mathematical distance or not is often insignificant for the final evaluations.'' Again, it would be helpful if the authors could provide an example of where a failure to consider whether a metric is a distance or not could cause a user to be misled.

The section on interpretability is confusing, in that it's not clear what Definition 11 or the accompanying discussion have to do with interpretability. I think this section requires more discussion to be useful.

A broad issue is that many of the definitions, terms, claims, and premises are ambiguous or underspecified. Some examples:
- The ontology on the right hand side of Figure 1 is unclear. The terms in the four leaf nodes are not clearly defined. Furthermore, the text below this figure mentions "four main groups" of metrics, but where do these four groups fit into the tree in Figure 1? Furthermore, these four group names don't appear in the text again, so I have no idea what they refer to.
- Figure 1: what is the base permutation used in the heatmap? Or is the disagreement ratio somehow insensitive to the base permutation? That doesn't seem intuitively likely, but if it's the case then this needs to be explained.
- Section 4.1, definition of a metric. They define a metric as taking two permutations as input but note that "all the given definitions work correspondingly for one-input metrics." I don't see how that's possible when the subsequent definitions involve two inputs, so I don't know what this means.
- Definition 2: What is $\mathcal{P}(S_n)$? It's not defined anywhere. Is this a typo and it should just be $S_n$? If it is meant to be $S_n$, then doesn't $S_n$ grow factorially rather than exponentially?
- Definition 2: Again, wouldn't this be highly sensitive to the base $\sigma \in S_n$? Some discussion of this is needed.
- Why is symmetry "impossible" in metrics for recommender systems and information retrieval?
- Definition 5: doesn't $\epsilon$ need to appear in the definition? Or are they fixing $\epsilon$ somehow?
- Definition 6: Shouldn't this be for all $\mu \in S_n$ as well?
- Definition 7: the width of a swap should be defined before it's used. What does "different impact" mean precisely? And does "evaluation" here just mean "metric"?
- Definition 8: What does "evaluates... equally" mean?
- Definition 9: I don't understand what is meant by $\lim_{k \rightarrow n} \epsilon_k = 0$. If $\epsilon_n = 0$, then this is satisfied trivially. However, $m_{@ k+1}$ is undefined for $k = n$, so this doesn't seem to be well-formed.
- Definition 11: What is $m_\text{max}$? I can make some guesses, but there are multiple possibilities. Additionally, are $m_\text{max}$ and $m_\text{min}$ defined conditional on $n, S_n$, or only on one or the other, or on neither? This is another place where some examples would really help.
- Below Definition 11: 1 and 2 seem to be redundant.
- There needs to be a clearer distinction between the mathematical definitions of the 8 properties and procedures for estimating whether these properties hold. For example, the caption for Figure 1 should state the parameters used to estimate these agreement ratios.

---

### Decision · Action_Editor_WeoV · 2024-08-08

**Recommendation:** Reject

**Comment:**

Ultimately, all three reviews leaned towards rejection. The reviews identified multiple concerns, and ultimately reviewers felt some of them were not sufficiently addressed by the authors.

The reviewers did appreciate several aspects of the paper:

* The paper was considered to have useful potential as a survey [4oAK]

* Unifying analysis of different metrics was appreciated [h2Jp,QFtL] and considered helpful for choosing metrics [QFtL]

* The mathematical framework was considered clearn, straightforward and comprehensive [h2Jp]

* The derived properties [h2Jp] were appreciated and considered well-motivated [QFtL], and detailing which properties each metric satisfies [4oAK,QFtL] was appreciated

* discussion of how the properties correlate with different applications was appreciated [QFtL].


However, several issues were raised, some of which remained a concern even after discussion with authors:

* Lack of clarity of the writing was criticized [QFtL]

* Use case of the framework for a downstream user was considered unclear [h2Jp]

* Lack of discussion on the choice of the eight properties was criticized, and how not considering them could cause misuse [h2Jp]

* The scope of what kinds of metrics are considered was considered unclear [h2Jp]

* Lack of clarity of Section 3 and its integration with appendix B was criticized [4oAK,QFtL]

* The value of the distance considerations was considered unclear [h2Jp]

* The interpretability discussion was considered unclear [h2Jp]

* Many definitions and claims were considered ambiguous [h2Jp,4oAK,QFtL] or unjustified [4oAK]

* More discussion of related work was desired [4oAK], and clarification of the novelty of the current work [QFtL]

* More discussion of the importance of the metric properties was desired [QFtL], as well as a mapping from domain areas to their important properties and suggested metrics [QFtL]

Although the authors responded to reviews and provided some clarificationa and improvements, reviewers felt concerns of clarity and completeness [QFtL] were not sufficiently addressed. One reviewer considered that re-evaluating the clarity/presentation issues might necessitate a full second review round [4oAK]. Particular remaining issues included discussion of a symmetry property [h2Jp] and how the definitions work for a one-argument metric [h2Jp], and relevance of metric ontologies [h2Jp].

Clarity of the evidence supporting claims of the paper is one of the TMLR acceptance criteria, and it seems the current submission is not yet at a ready state in this aspect, thus I must agree with the reviewers and recommend rejection.

**Audience:**

The reviewers indicated the paper could be useful to practitioners and both as a survey and to help choosing metrics. See "Comment" for details.

**Claims And Evidence:**

The clarity of the paper was considered insufficient by the reviewers along several aspects. See "Comment" for details.

**Resubmission Of Major Revision:**

The authors may consider submitting a major revision at a later time.